# Classification and Prediction of Natural Streamflow Regimes in Arid Regions of the USA

**Angela M. Merritt** [1,*,†], **Belize Lane** [2]  and **Charles P. Hawkins** [1,*] 

1   Department of Watershed Sciences, National Aquatic Monitoring Center, and Ecology Center,
    Utah State University, Logan, UT 84322-5210, USA
2   Department of Civil and Environmental Engineering, Utah State University, Logan, UT 84322-5210, USA;
    belize.lane@usu.edu
*   Correspondence: angiem.merritt@gmail.com (A.M.M.); chuck.hawkins@usu.edu (C.P.H.)
†   Present address: United States Bureau of Land Management, Prineville District Office, 3050 NE 3rd St,
    Prineville, OR 97754 USA; angiem.merritt@gmail.com.

**Abstract:** Understanding how natural variation in flow regimes influences stream ecosystem structure and function is critical to the development of effective stream management policies. Spatial variation in flow regimes among streams is reasonably well understood for streams in mesic regions, but a more robust characterization of flow regimes in arid regions is needed, especially to support biological monitoring and assessment programs. In this paper, we used long-term (41 years) records of mean daily streamflow from 287 stream reaches in the arid and semi-arid western USA to develop and compare several alternative flow-regime classifications. We also evaluated how accurately we could predict the flow-regime classes of ungauged reaches. Over the 41-year record examined (water years 1972–2013), the gauged reaches varied continuously from always having flow > zero to seldom having flow. We predicted ephemeral and perennial reaches with less error than reaches with an intermediate number of zero-flow days or years. We illustrate application of our approach by predicting the flow-regime classes at ungauged reaches in Arizona, USA. Maps based on these predictions were generally consistent with qualitative expectations of how flow regimes vary spatially across Arizona. These results represent a promising step toward more effective assessment and management of streams in arid regions.

**Keywords:** flow regimes; arid regions; classification; predictive models; nonperennial; ephemeral; perennial





## 1. Introduction

The effectiveness of the biological indices used to quantify the ecological condition of stream ecosystems depends on how accurately and precisely we can predict the reference condition for individual stream reaches, i.e., how biological communities and ecological processes naturally vary among reaches [1–3]. Accurate and precise predictions of the reference condition allow us to express assessments of biological condition relative to each site's specific biological potential and thus minimize confounding effects of natural variation in aquatic life with effects of human-caused environmental alterations [2]. In general, we know that naturally occurring differences in annual and interannual streamflow patterns and spatial intermittency along stream channels can be important drivers of variation in aquatic biodiversity and ecological processes [4–10], but this knowledge has not been fully used during the development, application, and interpretation of biological indices [3–6,11–13]. We expect that estimates of site-specific reference conditions will be especially critical in regions where flow regimes can vary markedly among stream reaches. One qualitatively conspicuous difference in flow regimes is whether flows are perennial or nonperennial. Unfortunately, our understanding of the physical, chemical, and biological variation that occurs among nonperennial reaches lags behind our understanding of perennial reaches even though nonperennial reaches often dominate in many regions [12–17].

In this study, our goal was to produce a better understanding of the similarities and differences in flow regimes that exist among both perennial and nonperennial reaches in the arid western United States. Our focus was to quantify major patterns of seasonal flow within large regions. We recognize that flow can also vary along individual streams, but quantifying that level of variation was beyond the scope of this study.

In general, the aquatic species that inhabit nonperennial reaches are often distinct from those that inhabit perennial reaches [18–22]. Moreover, reaches in arid regions can be especially critical to overall landscape health compared with their mesic counterparts [12–18,22–27]. However, it is not yet clear how diverse or predictable nonperennial reaches are from one another in terms of either their hydrologic regimes [12,14–17,23–26] or the biota they support [12,13,18–22,25,27]. Water resource managers need to more fully characterize the diversity of flow regimes that occur in these regions, and they need to be able to predict and map where they occur [23–26,28–30]. Our lack of understanding of the hydrologic heterogeneity that exists across stream reaches in arid regions, and the extent to which that heterogeneity is linked to naturally occurring variation in biota, currently limits the development and application of robust bioassessment tools for these reaches [4,13,21,27,31,32]. Accurately classifying the hydrological regimes of the 1000s of ungauged reaches in arid regions is a critical precursor to the development and application of bioassessment programs in these regions [16,28,31–34].

Perhaps the most striking difference between reaches in arid and mesic regions is the extent to which arid-region reaches experience drying. Over 90% of the reaches in an arid region may be nonperennial, and the degree to which reaches are nonperennial can vary greatly from one or a few days of zero surface flow per year to most days having no surface flow across numerous years [12–19,23–25,35]. Some nonperennial reaches, typically in mountainous arid regions, are nearly perennial, with few or no zero-flow days in at least some years and often exhibit seasonal snowmelt or storm-driven streamflow pulses [6,23–25,34–39]. The flow regimes of these nearly perennial reaches are influenced by differences in elevation and resulting differences in the type and timing of precipitation [6,15–17,35,36], as observed in perennial reaches [39–44]. These nearly perennial regimes are also often influenced by more extreme seasonal differences in snow cover and extended periods of drought than observed in perennial reaches [6,15–17,23,24,39,45,46]. Other reaches in arid regions can have ephemeral flow regimes, characterized by short-duration, high-peak flow events interrupting extended periods of zero flow associated with very low or no baseflow throughout the year [14–17,33–38]. These flow regimes appear to be influenced by shallow bedrock, high evapotranspiration rates, variable drought cycles, and episodic rain events interrupting droughts [15,38–46].

Researchers and managers have previously classified streamflow regimes into discrete groupings as a way to generalize about how reaches differ in their flow characteristics, quantify their diversity, and predict biotic responses to flow [6–8,40–44]. Most previous classifications have used hierarchical cluster analysis to identify classes. These analyses are ideally based on a large number of sites with continuous, long-term (>20 years) records that robustly characterize observed daily streamflow patterns through space and time [40–44]. Unfortunately, arid regions generally have limited gauge records, and those reaches that are gauged often have short or incomplete records [17,47–49]. Use of short-term or incomplete records can result in inaccurate and unrepresentative characterization of the actual flow regimes that exist. These types of data limitations can introduce large uncertainties in both flow-regime classifications [17,47,50] and their prediction [17,28]. These data limitations have compromised the extent to which we have been able to accurately characterize reference conditions for reaches in semi-arid and arid regions [16,17,28,33,34].

Even if an accurate classification of the variety of flow regimes that occur in arid-region streams existed, predicting what class ungauged reaches belong to may be more challenging in arid regions than mesic regions because of our incomplete understanding of some of the physicoclimatic controls on stream flows in arid regions. Some watershed attributes previously used to predict perennial streamflow regimes [40–44] may be less

influential in nonperennial reaches [6,17,29,30,45]. For example, some of the watershed attributes used in Dhungel et al. [41] to predict flow-regime classes in the conterminous United States included watershed area, mean elevation of the watershed, the difference between mean annual maximum and mean annual minimum precipitation, percentage of precipitation that occurs when mean air temperature is less than zero, proportion of annual precipitation that occurs in June, proportion of annual precipitation that occurs in October, and soil permeability. Additional physicoclimatic attributes, such as evapotranspiration [6,23,45–47], snow cover [51], or streambed topography [34,52–54], may be needed to better predict the occurrence and frequency of zero-flow conditions [17,36–39]. Furthermore, low correlation and nonlinear dependencies between rainfall and runoff in arid regions [17,45,55] may fundamentally constrain our ability to predict flow regimes of arid-region reaches.

These classification and prediction challenges are exemplified in the most arid regions of the southwestern USA. For example, only 25 US Geological Survey (USGS, Reston, VA, USA) reference-quality streamflow gauge records exist across Arizona [56]. Seventeen of these records start in the mid 1900s, but they have significant gaps or cease completely after 15–20 years, most notably after 1985 [57]. Furthermore, these streamflow records are especially sparse and incomplete for nonperennial reaches [57,58], a pattern that is true for most extremely arid regions [4,12,23]. The limited availability of streamflow records in the southwestern USA presents a specific challenge for both the identification of flow-regime classes and their spatial prediction [28]. Here, we present an approach to augment the spatial and temporal availability of streamflow data in an arid region with limited gauges and then use these extended streamflow records to develop and evaluate (1) alternative streamflow-regime classifications applicable to the arid southwestern USA and (2) empirical models to predict the streamflow regimes of ungauged reaches from publicly available geospatial data.

## 2. Materials and Methods

### 2.1. General Approach

The workflow required to complete analyses consisted of several steps (Figure 1), which we summarize here. In Section 2.2, Section 2.3, Section 2.4, Section 2.5, Section 2.6, Section 2.7, Section 2.8, Section 2.9, we describe each step in more detail. We first identified gauges from the USGS GAGES II [56] dataset of reference-quality (i.e., minimally-impeded flows) reaches that occurred in two arid southwestern states (Arizona and New Mexico) and had flow records between water years 1973 and 2013 (41 yr). USGS water years start on 1 October and end on 30 September0. For the purpose of this study, we defined reaches with one or more zero-flow days within the record of 41 years as nonperennial [16,17,58]. However, too few gauged nonperennial reaches with full records occurred in these two states to support a robust analysis. To increase the number of nonperennial reaches with full records, we extended the study area northward and eastward into the arid and semi-arid regions of surrounding states (Figure 2) until we had identified 45 gauges on nonperennial reaches that had full records. We then selected 45 gauged perennial reaches with full records that spanned the same geographic area as the nonperennial reaches. We also identified gauged reaches within the expanded study area that had abbreviated or discontinuous daily flow data for the same time period ($n = 197$). To estimate what the full flow records would have been at these 197 reaches, we developed random forest (RF) regression models in R [59–61] to predict the missing daily flow values in each of the 197 partial flow record from the mean daily flows observed at the 90 gauges with full records (see details below). These model-generated, synthetic streamflow data were then used to extract a variety of site-specific flow metrics or statistics for each reach. These metrics represented components of either streamflow magnitude, frequency, duration, timing, and rate of change [7,40–44] or components of the frequency, duration, and timing of zero flows [17,58]. We then used hierarchical cluster analysis to identify groups of reaches that differed in one or more flow metrics. We also calculated the number of zero-flow days that occurred in each record

and summarized these data as (1) the portion of a record with zero-flow days and (2) the portion of the 41 water years containing one or more zero-flow days. From these summary data, we developed additional classification schemes based on different threshold values of (1) the proportion of zero-flow days (ZFD) and (2) the proportion of years with one or more zero-flow days (ZFY). We then developed additional RF models to predict class membership for each of these classifications from watershed physicoclimatic data. We also used RF to model the continuous variation in percent ZFD and percent ZFY observed in the records.

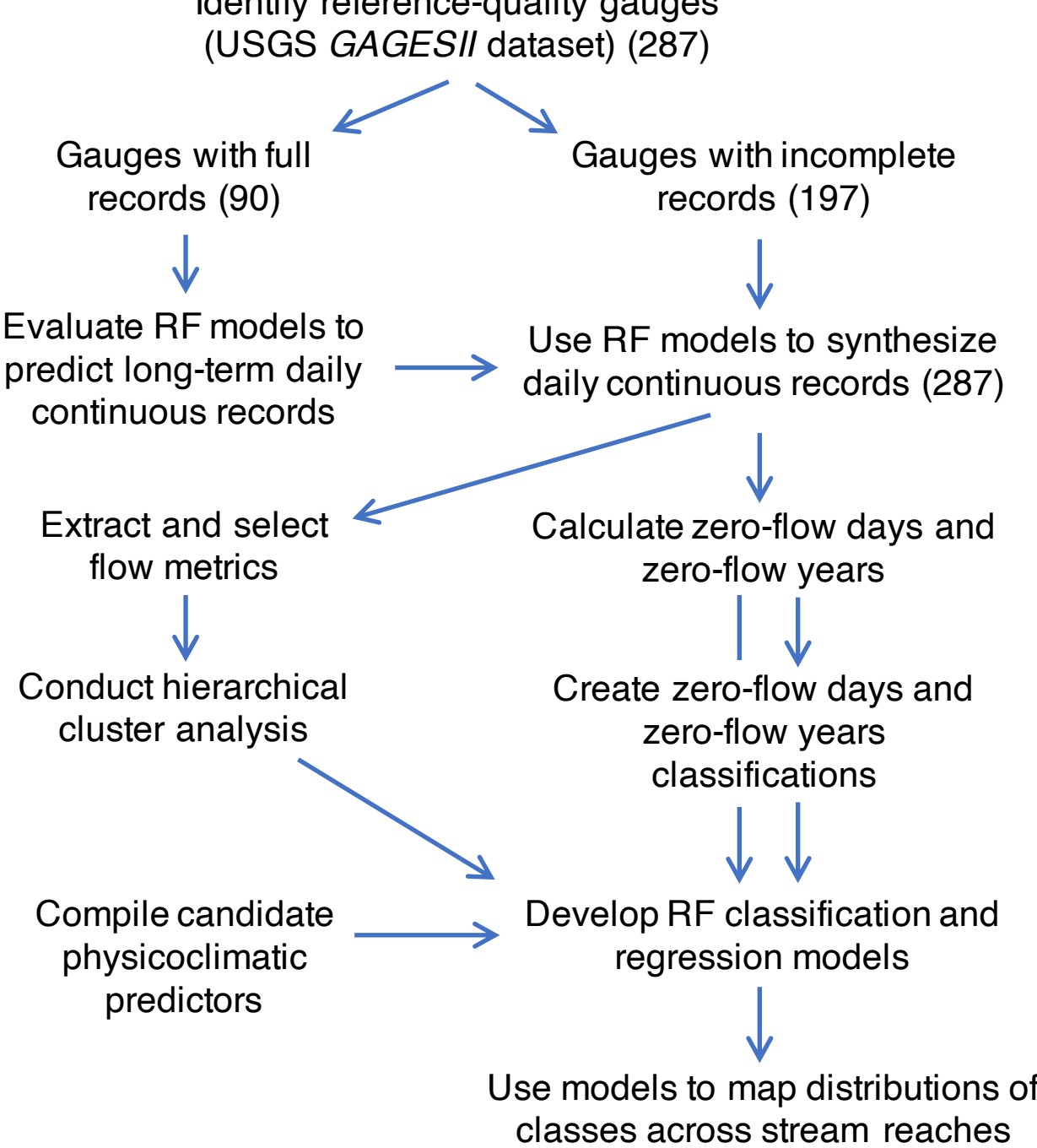

**Figure 1.** Workflow describing data compilation, pre-analysis data manipulation, classification, modeling, and mapping.

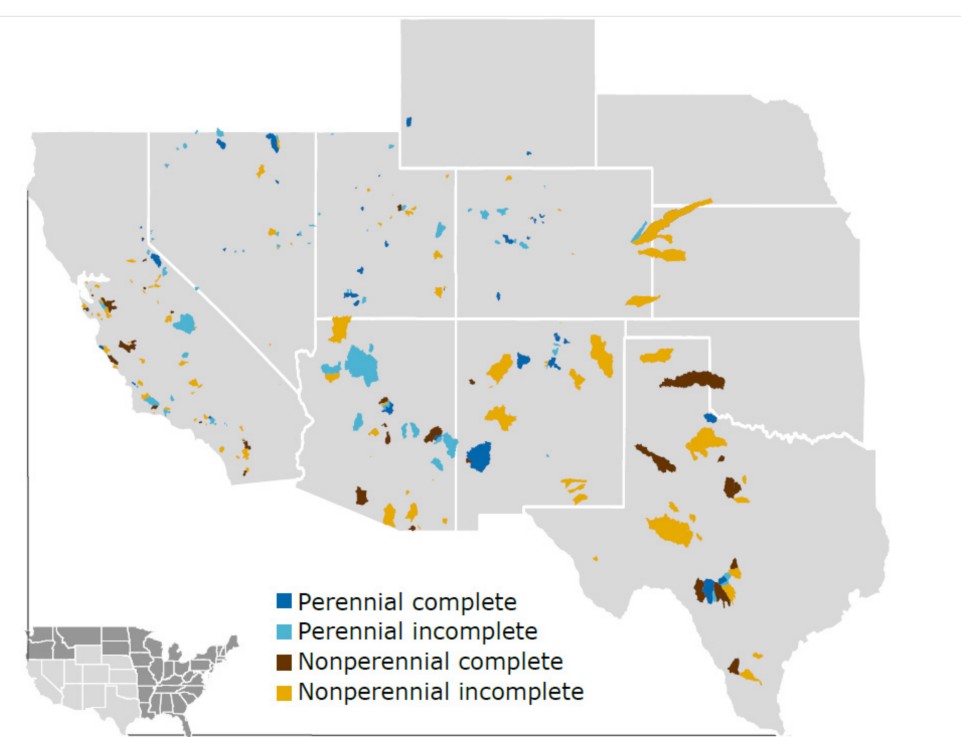

**Figure 2.** Map of the 287 basins with complete (dark blue, dark brown) and partial (light blue, light brown) USGS GAGES II reference streamflow data. Nonperennial reaches are in brown and perennial reaches are in blue.

### 2.2. Study Region

As a whole, the USA has one of the largest and longest datasets of gauged streamflow in the world [48,49]. The study region also has a relatively large number of gauged reaches with unimpeded flow relative to other arid regions in the world [62]. The gauged reaches used in this study spanned 22 Level III Ecoregions [63], with multiple records predating 1920 [57]. However, at least 6 of the coastal CA reaches we used may have gauge errors (too many zero-flow days) associated with flow reversals caused by tides [17].

### 2.3. Streamflow Data Collection

We used mean daily flow values instead of minimum instantaneous daily flow values to maximize comparability of our results with previous classifications of flow regimes. Of the 287 reaches used in this study, 90 reaches had 41 water-years of continuous mean daily flow measurements (1972–2013), and 197 reaches had between 22 (0.15%) and 14,611 (97%) days of missing data across the 14,975 days of record (Table 1). Forty-five of the 90 reaches with complete records were perennial (no zero-flow days in the period of record), and 45 reaches were nonperennial. Of the remaining 197 reaches with partial records, 122 were nonperennial with between 2% and over 99% zero-flow days (Table 1). Of all 167 nonperennial reaches, 80 had records with less than 20% zero-flow days and 87 reaches had records with more than 20% zero-flow days.

**Table 1.** Numbers of complete and partial streamflow records for each type (N = nonperennial and P = perennial) of reach used in this study, and the range in the amount of variation (pseudo-$r^2$ values) in mean daily streamflow explained by the RF models for each type of reach and record.

| Type of Record | Number | Variance Explained (%) * |
|:---:|:---:|:---:|
| Complete | 90 | 41.8–99.6 |
| NP | 45 | 41.8–99.5 |
| P | 45 | 80.0–99.6 |
| Partial | 197 | −2.3–99.6 |
| NP | 122 | −2.3–99.3 |
| P | 75 | 45.3–99.6 |
| Total | 287 | −2.3–99.6 |
| NP | 167 | −2.3–99.5 |
| P | 120 | 45.3–99.6 |

* Pseudo-$r^2$ values can be <0 but imply that no variation in daily flows were associated with the predictors.

### 2.4. Creating Synthetic Complete Streamflow Records for All Gauged Reaches

We developed random forest (RF) models [59–61] to predict missing mean daily flow values at the 197 reaches with incomplete data from the flows observed at the 90 reaches with full data. We first evaluated how well RF models could predict known flows by modeling mean daily flows at each of the 90 reaches with full data as a function of the flows observed at the other 89 reaches with full records. These 90 models predicted mean daily flows with pseudo-$r^2$ values ranging from 0.42 to > 0.99 (Table 1). Only 3 of these 90 models explained less than 80% of the variation in observed values. We therefore built RF models to predict missing flow values in the records of reaches with partial data. These models were calibrated with the existing observations in each partial record, and then the models were used to fill in the missing values. These models had pseudo-$r^2$ values ranging from 0.00 to > 0.99 (Table 1). Across all 287 models, 240 models accounted for over 90% of the variation in observed daily flow values. Seventeen other models accounted for between 70% and 90% of the observed variation in mean daily flows, and the remaining 20 models accounted for less than 70% of the variation in daily flows. With one exception, the models with the lowest pseudo-$r^2$ values (−2 to 70%) were those for nonperennial reaches (Table 1). Of the 20 records for which models explained less than 70% of observed variance in mean daily flows, 19 were from nonperennial reaches with high percentages of zero-flow days. The exception was a perennial reach with nearly constant flow.

To avoid negative values in synthetic data (all regression models can predict negative values at low values because of prediction error), we adjusted predicted daily flow values to maintain the same proportion of zero-flow days as in the raw data for each reach. We first calculated the percentage of days with mean daily flow equal to zero based on the original data, regardless of whether records were complete or partial. We then identified the predicted daily flow value associated with this percentage for each reach record. Next, we identified all occurrences of mean daily flow at or below this record-specific threshold. Finally, for each predicted record, we replaced all daily flows at or below their record-specific threshold flow values with zero. This adjustment assumes that the observed proportion of zero-flow days is representative of the entire 14,975 day period, including missing dates. This assumption may be inaccurate in reaches where missing data are associated with an anomalously dry or wet period, and error is expected to increase in reaches with more extreme low flows and with a greater proportion of missing records. However, we expect that this decision had limited influence on the derivative flow metrics and classification because threshold flow values were always small (near zero). Process-based hydrologic models may possibly generate more accurate daily streamflow timeseries over missing periods, but such models require substantial data inputs and parameterization, cannot be directly applied to other reaches, and often perform poorly under low-flow conditions. These issues remain critical limitations for application to large, data-limited regions [50].

### 2.5. Selection of Flow Metrics for Use in Hierarchical Classification

Flow-regime classifications are typically based on several flow metrics that together characterize different aspects of streamflow magnitude, frequency, duration, timing, and rate of change [7,40–44]. Dhungel et al. [41] used principal components analysis to identify 16 candidate flow variables from 65 metrics that characterized broadly different dimensions of these five flow attributes. We used identical methods to calculate site-specific values for the same 16 metrics, but we considered 6 other flow metrics related to zero-flow days instead of the two low-flow metrics used by Dhungel et al. [41] because we wanted to provide greater resolution in distinguishing different types of nonperennial streamflow patterns. We examined correlations between these 20 candidate metrics and dropped one of any pair of metrics that were correlated ($r > |0.75|$) with one another. When selecting which metric to use from a pair of correlated metrics, we used the one that we considered most likely to have an interpretable effect on aquatic life [5,8,18–22]. We ultimately selected 12 metrics for use in the hierarchical cluster analysis (Table 2). The final 12 metrics included 9 used by Dhungel et al. [41] and 3 metrics characterizing different aspects of zero-flow conditions: the mean number of zero-flow days per year and its coefficient of variation (ZcntMn, ZcntMnCV) and the mean duration of continuous zero-flow days across the period of record (Meanzd_R).

**Table 2.** The 12 flow metrics used in the hierarchical cluster analysis. Abbr = abbreviation.

| Flow Metric Description | Abbr |
|---|---|
| 1. Mean number of zero-flow days per year | ZcntMn |
| 2. Coefficient of variation of the mean number of zero-flow days per year | ZcntMnCV |
| 3. Seven day moving average of minimum flow | Q7min |
| 4. Bank full flow | BFF |
| 5. Flood duration (mean number of days exceeding bank full flow) | FD |
| 6. Mean days to annual peak flow | Pk_time |
| 7. Mean duration of all zero-flow events across the record | Meanzd_R |
| 8. Mean days to 50% of total annual flow | T50 |
| 9. Mean number of low-flow events per year | LFE |
| 10. Mean number of high-flow events per year | HFE |
| 11. Constancy: a unitless measure of uncertainty, higher C = high certainty throughout year | C |
| 12. Contingency: a unitless measure of seasonal uncertainty, higher M = high certainty by season | M |

Bank full flow was estimated as $Q_{1.67}$, which is the discharge value that has a probability of exceedance of 1/1.67 from a lognormal probability distribution fit to the annual maximum daily flow series.

### 2.6. Classifying Streamflow Regimes

Hierarchical cluster analysis has been frequently used to characterize streamflow regimes [41–44]. For our analyses, we used Ward's method of agglomerative hierarchical cluster analysis [59]. This method produces a dendrogram that describes similarities among sites based on the joint variation among sites in the metrics used in the classification. The dendrogram can then be visually inspected to identify increasingly resolved and subtle differences among classes as the number of branches increases [41–43]. We were primarily interested in how well 3–5 classes partitioned variation in flow regimes, which represented a trade-off between resolution (which will affect the accuracy and precision of biological assessments), the accuracy in predicting class membership of ungauged reaches, and the practical needs of water resource managers for a classification scheme that can be easily communicated to stakeholders.

We also created 4 alternate classifications based directly on the percent of zero-flow values in the streamflow records for comparison with the cluster-based approaches (Table 1). This independent threshold-based approach has been previously used to classify non-perennial streamflow regimes in the arid southwestern USA [28]. For this approach, we calculated both percent of zero-flow days (ZFD) across the record for each reach as well as the percent of years across each reach's record with at least one zero-flow day (ZFY). As discussed above, we assumed that measures of ZFD and ZFY were not affected by differ-

ences among reaches in the number and distribution of missing data in the partial-record reaches. We then applied different thresholds to define three or four different classes for both the ZFD and ZFY classifications. These thresholds were selected to create classes that generally aligned with the flow characterization methods being considered to support of bioassessments of southwestern USA reaches [16,28], which we also thought could be (1) biologically relevant and (2) predictable from watershed attributes. ZFD thresholds were set at 0% (the 120 perennial reaches), >0%, >2%, and >20% ZFD for a four-group classification and at 0%, >0%, and >20% for a three-group ZFD classification. Thresholds for the ZFY classifications were set to create two three-group classifications with class thresholds of 0% ZFY (the same 120 perennial reaches), >0%, and >20% ZFY and another with 0% (the same 120 perennial reaches), >0%, and >75% ZFY thresholds.

We also used RF regression models to assess if continuous variation in percent ZFY and percent ZFD was predictably associated with variation in physicoclimatic conditions across the study region. The rationale for these analyses was that if the models accounted for a substantial portion of the variation in ZFY and ZFD, managers could use them to map flow regimes based on any threshold of ZFY or ZFD that was appropriate to their specific management needs.

*2.7. Selection of Predictor Variables*

We considered over 500 landscape and climate attributes as candidate predictor variables of both the streamflow-regime classes and the continuous variation in ZFDs and ZFYs. Seventy-one of these attributes were obtained from the USEPA's (United States Environmental Protection Agency, Office of Water, Washington, DC, USA) StreamCat dataset [64], which was built on the NHDPlusV2 geospatial framework. StreamCat includes geographic descriptors such as latitude, longitude, and watershed area as well as watershed-level attributes describing climate, vegetation, soil, and geomorphology, many of which are known to be associated with flow generation and variation in the magnitude, frequency, duration, timing, or rate of change of streamflow [7,40–44]. For example, StreamCat includes a topographic wetness index and a soil wetness index, both of which have been used as predictors of runoff in hydrologic models [65]. We expected these watershed- and local catchment-level attributes to be important predictors of flow-regime classes, particularly the base flow index (BFI) [12,15,38]. The StreamCat BFI was generated through spatial interpolation of USGS gauge-specific BFI calculations, where BFI represents the baseflow volume/total streamflow volume as calculated through a combination of a moving minimum flow (similar to the Q7min in Table 2) and a recession slope test [66,67]. StreamCat data are available at two scales: local catchments and full-contributing watersheds, but our variable selection procedure almost always selected predictors that were calculated for the entire full-contributing watershed above the downstream boundaries of the NHDPlusV2 reaches on which gauges were located. The use of NHDPlus-associated predictors limited the universe of stream reaches to which we could make predictions, i.e., the models cannot make predictions to small headwater reaches that are not included in the NHDPlus.

Given the driving role of evapotranspiration and climate on streamflow patterns in arid landscapes [23,39,45,46], several additional evapotranspiration and aridity indices not available in StreamCat were calculated from 30 year, 4 km (1981–2010) mean air temperature and precipitation PRISM (Parameter-elevation Regression on Independent Slopes Model, prism.oregonstate.edu) data [68] and the modeled runoff index in StreamCat [64]. These variables were combined through simple algebra [69]:

Long-term evapotranspiration (*ET*, mm) was calculated from PRISM precipitation (Precip8110Ws) [64,68] and StreamCat Runoff (RunoffWs) [64] as:

$$ETmm = Precip8110Ws - RunoffWS \tag{1}$$

We followed Gardner et al. [70] to estimate regional potential evapotranspiration (*PET*) from PRISM mean annual temperatures (Tmean8110Ws) [64,68] as:

$$PET = (1.2 \times 10^{10}) \times exp(-4620/(Tmean8110Ws +273)) \qquad (2)$$

We also followed Gardner et al. [70] to calculate an Aridity Index (PET_P) as:

$$PET\_P = PET/Precip8110Ws \qquad (3)$$

and an Evaporative Index (ET_P) as:

$$ET\_P = ETmm/Precip8110Ws \qquad (4)$$

We compiled additional watershed-level summaries of several climate variables [45,46,51] including the average watershed snow-cover from the Modis-10A1 V6 Snow Cover Daily Global 500 m product [71] and a county level drought-severity index from the National Drought Mitigation Center [72].

We used ArcGIS (ESRI, Redlands, CA, USA) [73] to generate curvature variables from a 30 m Digital Elevation Model (DEM) downloaded through Earth Explorer [74]. The *ArcGIS* curvature tool was used to calculate the second derivative of the DEM to generate planar and profile curvature rasters. Slope curvature, describing the convexity or concavity of the terrain, can affect runoff [38,65] through groundwater access [38,75] and saturation rate of soil profiles in response to rainfall [15,34,65,76]. Minimum, maximum, mean, and standard deviation of the perpendicular and parallel terrain curvature were used as additional candidate predictors.

We selected 95 of the >500 variables as candidate predictors (Table S1, Supplemental Materials). Predictors were eliminated if they lacked evidence of being associated with either runoff generation, base-flow contribution, or zero-flow events. We also removed predictors that were functionally redundant but calculated differently. In general, we retained the predictors that were most easily calculated to facilitate repeatability in future analyses. We also removed predictors that were missing from one or more of the 287 gaged locations or that varied little across the study region.

We then used the *VSURF* function in *R* [59,77] as an objective way to identify the most parsimonious set of predictors that produced the best model performance. *VSURF* uses RF to apply a 3-step variable selection process to further eliminate redundancy and identify the subset of predictor variables that produce the lowest out-of-bag errors for each model. *VSURF* will optimize variable selection based on either interpretation or prediction, and we chose to optimize predictive accuracy.

*2.8. Predicting Streamflow-Regime Classes and Continuous Variation in ZFDs and ZFYs*

We used RF models to predict both class membership and the observed continuous variation in ZFDs and ZFYs. RF models frequently perform better than many alternative types of classifiers and regression models [59–61]. They are especially useful when there are many predictor variables. RF models are also resistant to overfitting and automatically incorporate interactions between predictor variables. RF models create hundreds of classification (or regression) trees based on bootstrapped subsamples of the data and assess model performance based on the overall out-of-bag (OOB, observations withheld when building each tree) prediction error. For each classification model, we randomly selected *n* = the smallest class size by applying the 'sampsize' and 'strata' arguments in the RF algorithm. Balancing class sizes avoids inherent biases in over-predicting classes with larger numbers of observations [78]. We used confusion matrices to evaluate the prediction errors associated with each individual class in each classification. We used pseudo-$r^2$ values to evaluate the amount of variance RF regression models explained in observed ZFY and ZFD values.

### 2.9. Mapping Flow-Regime Classes

As a case study, we used ArcMap to predict the flow regimes of the ~75,000 NHD reaches in Arizona. The NHD (National Hydrography Dataset is a nationally-consistent, digital, 1:100,000-scale map of the stream and river network in the USA (https://www.usgs.gov/core-science-systems/ngp/national-hydrography). Reaches in the NHD are defined as the stream segments between two confluences and flows are assumed to be similar within a reach. We used local catchments (the hillslopes draining directly into each reach) as mapping units rather than reaches to help visualize predictions.

## 3. Results

### 3.1. Hierarchical Classifications

We identified a total of five hierarchical levels of flow regimes from the hierarchical cluster analysis that produced two, three, four, and seven nested classes of flow regimes (Figure 3, Table 3). Class A reaches included the driest reaches in the dataset and exhibited intermittent flood peaks associated with monsoonal rainfall (Figures 4 and 5). This class had the absolute highest mean annual count of zero-flow days (ZcntMn, Figure 4), which approached 300 days for A1 reaches and ~150 days for A2 reaches. In comparison, Class B reaches had ~75 zero-flow days. Class A reaches ranged from being largely ephemeral with over 75% zero flow and no baseflow (A1) to exhibiting monsoonal flow signatures (A2) with between 20 and 75% zero-flow days and some baseflow (Figure 5). Class A reaches had flood peaks that were orders of magnitude higher and Q7min values that were several orders of magnitude lower than in Class B. Some of these peaks were 30 orders of magnitude higher than any other reach (Figure 5), but baseflows in Class A reaches were at or near zero (Figure 4). Class A reaches also had a greater number of both high- and low-flow events (HFE and LFE) (Figure 4). Most Class A1 reaches with over 75% zero-flow days lacked seasonality, as quantified by high constancy (C) and low contingency (M) of flow (Figure 5). Collectively these metrics describe reaches with more extreme differences between high- and low-flow events than Class B reaches (Figures 4 and 5).

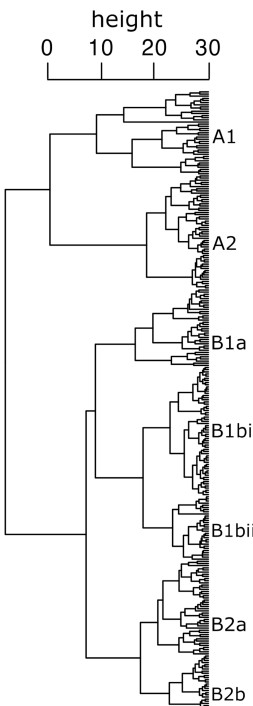

**Figure 3.** The dendrogram produced by the hierarchical cluster analysis with the most resolved seven-group classes shown. The number of reaches in each class were A1 (39), A2 (53), B1a (36), B1bi (60), B1bii (30), B2a (44), and B2b (25).

**Table 3.** The percent of nonperennial (NP) reaches in each hierarchical class and general descriptions of the flow regimes of the seven-group classification. P = perennial.

| Class | %NP | Class | %NP | Class | %NP | Class | %NP | Class | %NP | Description |
|-------|-----|-------|-----|-------|-----|-------|-----|-------|-----|-------------|
| A | 100 | A1 | 100 | A1 | 100 | A1 | 100 | A1 | 100 | Aseasonal-NP Ephemeral and flashy |
| | | A2 | 100 | A2 | 100 | A2 | 100 | A2 | 100 | Weak seasonal-NP Wet winter-spring |
| B | 38 | B | 38 | | | B1a | 97 | B1a | 97 | Seasonal-NP Rain and snowmelt Summer dry |
| | | | | B1 | 28 | B1b | 0 | B1bi | 0 | Weak seasonal-P Rain and snowmelt |
| | | | | | | | | B1bii | 0 | Seasonal-P Rain and snowmelt Summer dry |
| | | | | B2 | 58 | B2 | 58 | B2a | 91 | Aseasonal-NP |
| | | | | | | | | B2b | 0 | Aseasonal-P |

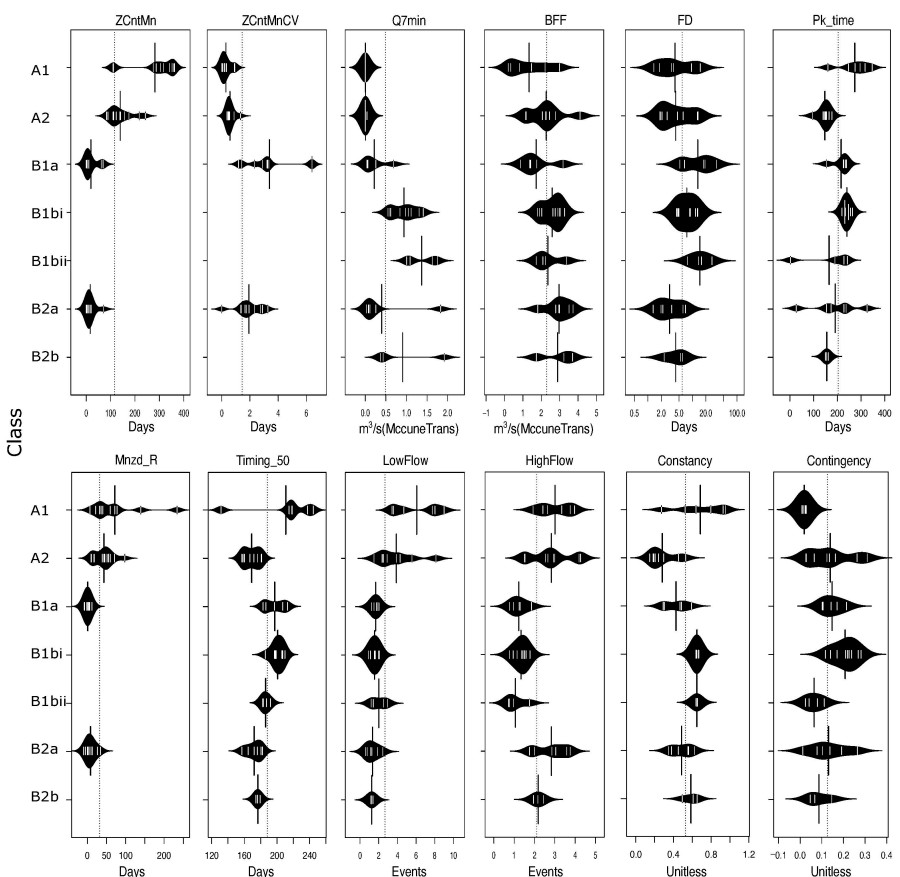

**Figure 4.** Density distribution plots of the metric values for each of the seven most resolved hierarchical classes (A1 to B2b). Plots for the different metrics are presented in the same order as given in Table 2. Each panel's dotted line represents the mean for all 287 streams. The bold intersecting line for each distribution represents the mean for each individual class.

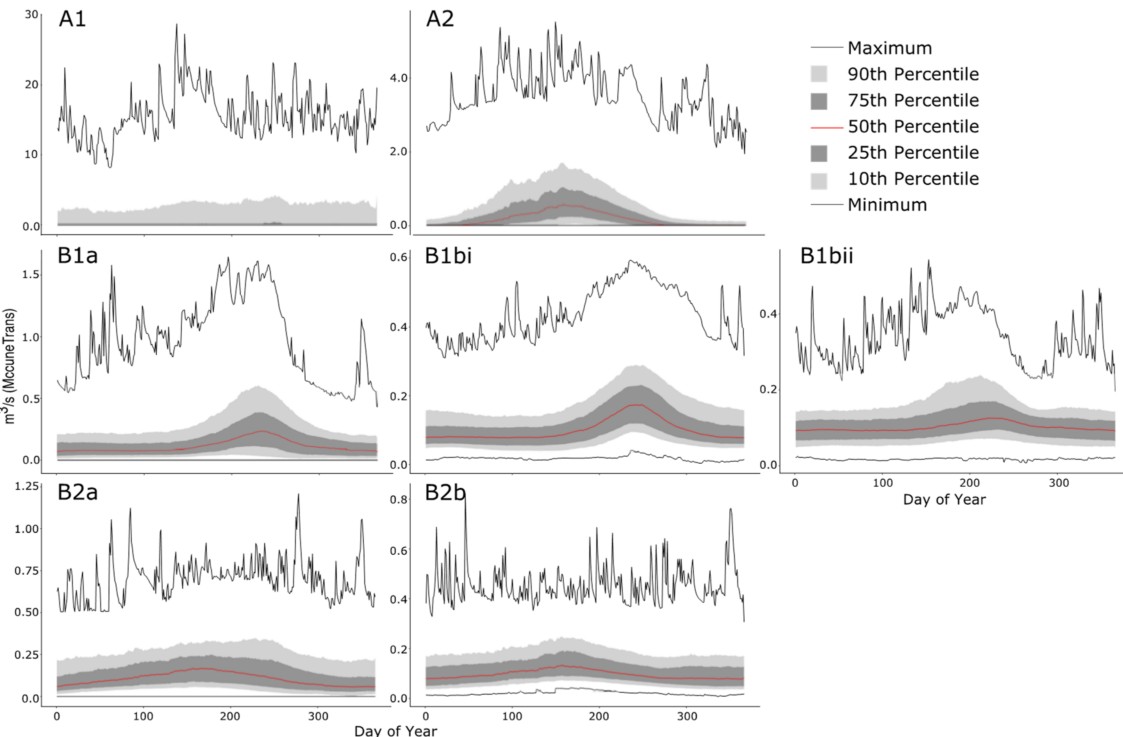

**Figure 5.** Dimensionless reference hydrographs (archetypes) for the seven most resolved classes (A1 to B2b). Day of year is based on water year, i.e., October 1 = day of year 1. The red line and shading trace the mean percentiles of mean daily flow values for each individual day of year calculated for each class across the full period of record. The black lines represent the maximum and minimum mean daily flows for each class on each day of record. The number of reaches in each class were A1 (39), A2 (53), B1a (36), B1bi (60), B1bii (30), B2a (44), and B2b (25).

Class B reaches included a mix of perennial and nonperennial reaches (Figure 3) characterized by seasonal variation in flow (Figures 4 and 5). This class was further split into reaches with snowmelt (B1) and rain storm (B2) signatures (Figure 5). Class B1 was further split into nonperennial (B1a) and perennial (B1b) reaches based on the presence of zero-flow days (Figures 3–5). Class B1a reaches were 97% nonperennial, with the highest ZcntMnCV values and some of the lowest mean zero-flow day durations (Meanzd_R) (Figure 4). The rain storm Class B2 was similarly split into nonperennial (B2a) and perennial (B2b) reaches (Figure 5).

### 3.2. Percent Zero-Flow Classifications

Both ZFY classifications included the same class of 120 perennial reaches by design but differed in the number and size of nonperennial classes that were assigned. One ZFY classification included 41 reaches ranging from >0 to 20% ZFYs and 126 reaches with >20% ZFY. The other ZFY classification included 83 reaches ranging from >0 to 75% ZFY and 84 reaches with >75% ZFY. The ZFY and ZFD classes were generally similar in the distribution of reaches among classes.

Both ZFD classifications also included a class consisting of the 120 perennial reaches but differed in the number and size of nonperennial classes. For the four-group ZFD classification, one group included 37 reaches with >0 to 2% ZFD, which were reaches predominantly from group B1a. Another class consisted of 43 reaches with >2 to 20% ZFD, which contained a mix of reaches from A2, B1a, and predominantly B2a. The alternative three-group ZFD classification combined these 37 and 43 (80) reaches into the >0 to 20% ZFD class. The last class for both ZFD classifications consisted of 87 reaches with >20% ZFD, which was almost exclusively class A reaches.

### 3.3. Model Performance

Overall OOB prediction error for the RF classification models varied from 15 to 41% (Tables 4 and 5), and pseudo-$r^2$ values varied from 50 to 58% for the two RF regression models. In general, and as expected, overall prediction error increased as the number of classes increased. Furthermore, class-specific prediction errors sometimes varied markedly, i.e., some types of flow regimes were more difficult to predict than other types. Performance of the models predicting the hierarchical-based reach classes varied both with the number of classes and among classes (Table 4). These models had overall OOB errors of 15, 23, 24, 34, and 39% for the two- through seven-group classifications, respectively. The variation between class-specific prediction errors became more pronounced as the number of groups increased. There were no obvious trends in what types of classes were predicted with greater or lesser accuracy. Overall model performance also varied across the ZFY and ZFD classifications (30–41% error) (Table 5). In general, the ZFY- and ZFD-based class models poorly predicted classes with intermediate values of ZFY or ZFD, exceeding 70% and sometimes 80% in class-specific prediction error. The results of the ZFY and ZFD RF regression models were consistent with results of the ZFY and ZFD threshold classification models (Figure 6). These two regression models explained slightly more than 50% of the variation in percent ZFY ($r^2 = 0.54$) and ZFD ($r^2 = 0.52$). However, the prediction errors were sometimes relatively large. For example, the prediction errors associated with any observed value could be as much as 30 or 40% of the actual values of ZFD and ZFY, respectively (Figure 6).

**Table 4.** Confusion matrices for the two- to seven-group hierarchical classification models showing model performance as out-of-bag (OOB) error. Classes with over 90% nonperennial reaches are marked with a *. Classes with 100% perennial reaches are marked with a †. Observed classes are rows, and predicted classes are columns.

| | A * | | B | | | Error (%) | Totals |
|---|---|---|---|---|---|---|---|
| | | | **Two classes (15% OOB)** | | | | |
| A * | 79 | | 12 | | | 13 | 92 |
| B | 32 | | 163 | | | 16 | 195 |

| | A1 * | A2 * | B | | | Error (%) | Totals |
|---|---|---|---|---|---|---|---|
| | | **Three classes (23% OOB)** | | | | | |
| A1 * | 32 | 4 | 2 | | | 16 | 39 |
| A2 * | 8 | 39 | 6 | | | 26 | 53 |
| B | 14 | 33 | 148 | | | 24 | 195 |

| | A1 * | A2 * | B1 | B2 | | Error (%) | Totals |
|---|---|---|---|---|---|---|---|
| | | | **Four classes (24% OOB)** | | | | |
| A1 * | 29 | 4 | 2 | 3 | | 24 | 39 |
| A2 * | 7 | 36 | 2 | 8 | | 32 | 53 |
| B1 | 2 | 5 | 108 | 11 | | 14 | 126 |
| B2 | 7 | 13 | 5 | 44 | | 36 | 69 |

| | A1 * | A2 * | B1a * | B1b † | B2 | Error (%) | Totals |
|---|---|---|---|---|---|---|---|
| | | | **Five classes (34% OOB)** | | | | |
| A1 * | 28 | 4 | 2 | 1 | 3 | 26 | 39 |
| A2 * | 5 | 33 | 3 | 2 | 10 | 38 | 53 |
| B1a * | 1 | 4 | 19 | 11 | 1 | 47 | 36 |
| B1b † | 1 | 5 | 13 | 68 | 3 | 24 | 90 |
| B2 | 6 | 16 | 1 | 4 | 42 | 39 | 69 |

| | A1 * | A2 * | B1a * | B1bi † | B1bii † | B2a * | B2b † | Error (%) | Totals |
|---|---|---|---|---|---|---|---|---|---|
| | | | | **Seven classes (39% OOB)** | | | | | |
| A1 * | 31 | 2 | 2 | 0 | 1 | 2 | 0 | 18 | 39 |
| A2 * | 6 | 34 | 1 | 1 | 5 | 2 | 4 | 36 | 53 |
| B1a * | 1 | 4 | 17 | 6 | 6 | 0 | 2 | 53 | 36 |
| B1b.i † | 0 | 1 | 6 | 51 | 1 | 1 | 0 | 15 | 60 |
| B1b.ii † | 1 | 2 | 2 | 7 | 15 | 0 | 3 | 50 | 30 |
| B2a * | 5 | 12 | 0 | 3 | 3 | 13 | 8 | 70 | 44 |
| B2b † | 0 | 4 | 0 | 0 | 2 | 5 | 14 | 44 | 25 |

**Table 5.** Confusion matrices for the four percent zero-flow day and percent zero-flow year threshold classification models. P = perennial reaches (equal to 0% ZFY or ZFD); NPy>0–20 = > 0 but <20% ZFY); NPy>20 = >20% ZFY); NPy>0–75 = >0 but <75% ZFY); NPy>75 = >75% ZFY); NPd>0–20 = >0 but <20% ZFD); NPd>20 = >20% ZFD); NPd>0–2 = >0, <2% (greater than zero, less than 2 percent ZFD); NPd>2–20 = >2 but <20% ZFD). Nonperennial ZFY classifications were based on thresholds first set at 20% ZFY and then at 75% ZFY for each of 287 reaches. Nonperennial ZFD classifications were based on thresholds first set at 20% ZFD and then at two higher thresholds: 2% and 20% ZFD. Model performance ranged from 24 to 37% OOB error. Observed classes are rows, and predicted classes are columns.

| | Zero-flow year (ZFY) thresholds | | | | |
|---|---|---|---|---|---|
| | 20% ZFY (31% OOB) | | | | |
| | P | NPy > 0–20 | NPy > 20 | Error (%) | Totals |
| P | 88 | 20 | 12 | 27 | 120 |
| NPy > 0–20 | 16 | 17 | 8 | 59 | 41 |
| | 16 | 17 | 93 | 26 | 126 |
| | 75% ZFY (30% OOB) | | | | |
| | P | NPy > 0–75 | NPy > 75 | Error (%) | Totals |
| P | 90 | 19 | 11 | 25 | 120 |
| NPy > 0–75 | 25 | 44 | 14 | 47 | 83 |
| NPy > 75 | 6 | 11 | 66 | 20 | 84 |
| | Zero-flow days (ZFD) thresholds | | | | |
| | 20% ZFD (34% OOB) | | | | |
| | P | NPd > 0–20 | NPd > 20 | Error (%) | Totals |
| P | 92 | 18 | 10 | 23 | 120 |
| NPd > 0–20 | 24 | 32 | 24 | 60 | 80 |
| NPd > 20 | 8 | 12 | 66 | 23 | 87 |

| | 2% and 20% ZFD (41% OOB) | | | | | |
|---|---|---|---|---|---|---|
| | P | NPd > 0–2 | NPd > 2–20 | NPd > 20 | Error (%) | Totals |
| P | 80 | 20 | 11 | 9 | 33 | 120 |
| NPd > 0–2 | 16 | 10 | 5 | 6 | 73 | 37 |
| NPd > 2–20 | 8 | 6 | 18 | 11 | 58 | 43 |
| NPd > 20 | 7 | 3 | 15 | 61 | 29 | 87 |

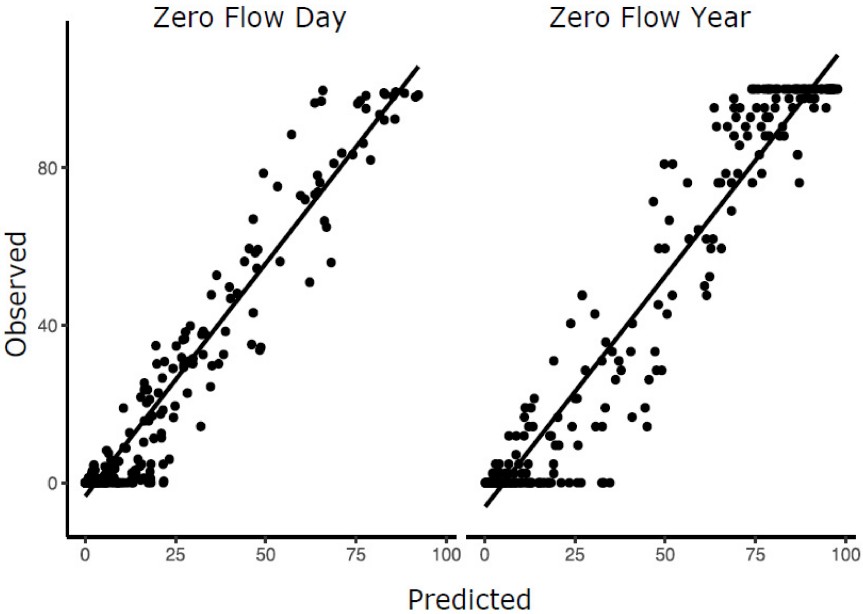

**Figure 6.** Plots of observed versus predicted values for the ZFY and ZFD regression models.

*3.4. Consistency in Selected Predictor Variables*

A few predictor variables were consistently selected from the 95 predictor variables across models (Table S2, Supplementary Materials). BFI was identified as the top predictor in all but the hierarchical four-class model, which ranked longitude before BFI. Other top predictors were related to location, topography, and climate. Other frequently selected predictors included measures of streambed shape (e.g., ParCurveSTD), air temperature (e.g., Tmax8110Ws), evapotranspiration (e.g., PET), and elevation (e.g., ElevWs). These predictors were also selected by the ZFY and ZFD models, along with additional dimensions of streambed shape and landcover.

*3.5. Flow-Regime Maps*

The different predicted hierarchical-based reach classes tended to be geographically organized across the state of Arizona (Figure 7). Most of the driest reaches (A1 and A2) were predicted to occur in the southwestern and northeastern parts of the state, whereas the most perennial reaches were predicted to be largely confined to two mountainous areas in the state: the most northwestern part of the state and the far east-central part of the state. In general, the driest classes transitioned geographically into less severely nonperennial reaches (class B1a), which transitioned into the mixed perennial/nonperennial class (B2), which transitioned into perennial reaches (class B1b). By mapping different levels of the hierarchical classification, the error inherent in modeling was apparent. For example, the specific reaches predicted to be in a certain class (e.g., A2) differed for some reaches depending on whether A2 was part of a three-, four-, or five-group classification. Although the predicted spatial distribution of each class of reaches was broadly similar regardless of the level of resolution used in the classification, the predicted class of specific reaches could differ.

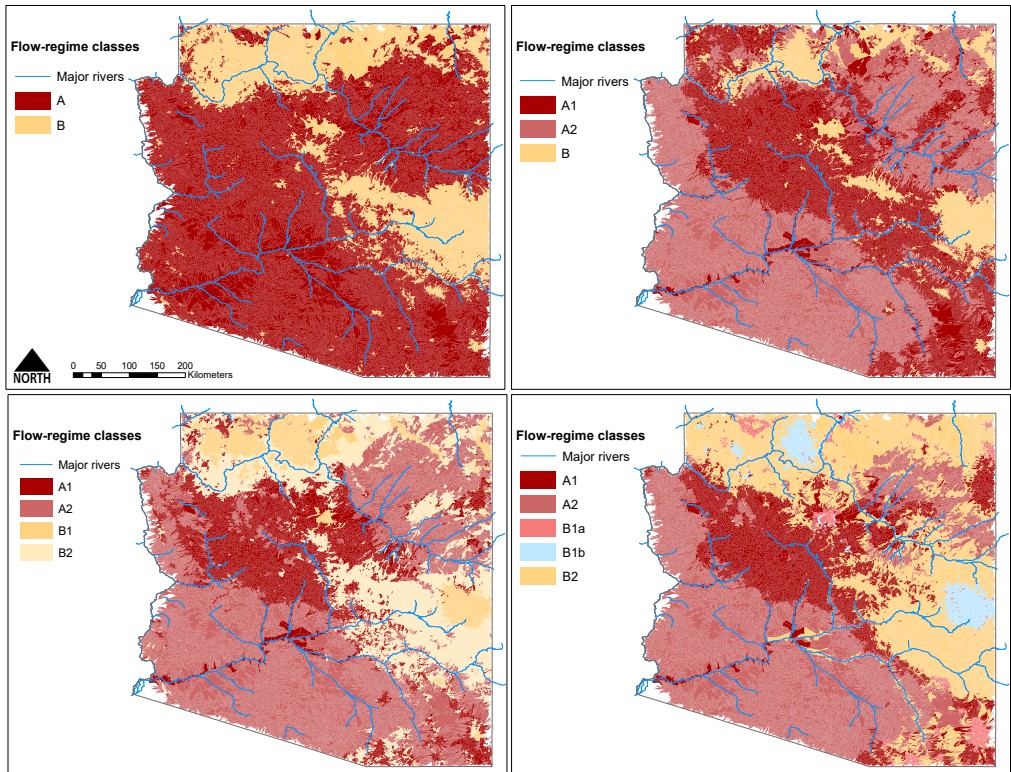

**Figure 7.** Maps of the state of Arizona showing the predicted locations of the three-, three-, four-, and five-group classes derived from the hierarchical cluster analysis. Major rivers (blue lines) are superimposed.

## 4. Discussion

Water resource managers currently lack the tools needed to assess the physical, chemical, and biological integrity of the majority of stream ecosystems in arid regions [4–6]. This problem is especially severe for nonperennial reaches, which are understudied relative to perennial reaches but which represent the vast majority of the stream networks in these regions [12–17]. Knowledge of how physicochemical factors structure biological communities provides the basis for defining and identifying the reference conditions on which assessments of individual waterbodies are based [1,3] and is thus central to successful assessment and management programs.

Of the many naturally occurring physicochemical factors that can influence aquatic life in streams, the flow regime is thought to be of paramount importance [5–10]. In arid regions, variation in flow regimes is likely to be especially critical to aquatic life given that a conspicuous dimension of these regimes is the extent to which flows completely cease—a naturally occurring disturbance that has profound effects on the type of biota that can persist in a reach [18–22]. Characterizing, or classifying, natural flow regimes in arid-region streams (and hence their reference conditions) is therefore a critical need in the USA and elsewhere [16,23–34]. Spatially sparse and temporally incomplete flow records typically limit our ability to robustly describe the variability of naturally occurring flow regimes in arid regions [17,23,47,48,58]. The lack of classification schemes beyond the traditional division of perennial and nonperennial reaches has likely limited our appreciation of the full diversity of ways that flow regimes may vary across these landscapes [4,24]. In this study, we first showed that we could improve the spatial coverage of daily streamflow data by estimating data that were missing from gauges with incomplete records. That process, together with the inclusion of additional flow metrics characterizing different aspects of zero flow, allowed us to explore several alternative classifications that may be of use to water resource managers. Below, we address both the potential utility of these classifications and factors that may limit their use, including our ability to predict where they occur.

### 4.1. The Value of Creating Standard, Long-Term Synthetic Flow Records

Developing robust classifications of flow regimes requires data that cover the full range of variability in flow. For arid regions, flow records should span a full range of both nonperennial and nonperennial reaches [12–17], and more than one metric related to the frequency, duration, and timing of zero flows are likely needed to describe differences in their flow regimes [6,17,24–26]. By using RF models to estimate the daily flows occurring at gauges with incomplete records from gauges with complete records, we greatly enhanced our ability to characterize the spatial and temporal diversity of stream flows that occur in the southwestern USA [17,28,47]. The majority (55%) of these models explained well over 90% of the variation in daily flows that were recorded in the 197 gauges with partial-records although, on average, flows in nonperennial reaches were predicted with slightly less accuracy than those in perennial reaches (Table 1). Previously used approaches to classify and predict flow regimes would have disregarded records from up to 145 gauges in our study region because they would not have met data-quality criteria, e.g., they either had less than 10–20 years of record, or they had data gaps greater than 5–10 years [47–49]. Furthermore, a majority of these 145 incomplete records fell within gauge-sparse arid ecoregions in Arizona, New Mexico, and Texas, 50 of which had been excluded by Dhungel et al. [41] in their analysis of the potential effect of climate change on flow regimes across the conterminous USA. By creating synthetic records, we greatly enhanced our ability to characterize variation across the region in all aspects of flow, especially those associated with zero flow (ZFY, ZFD).

### 4.2. Beyond the Perennial–Nonperennial Dichotomy

Water resource managers in the southwestern USA are starting to realize that a simple dichotomy of perennial and nonperennial reaches is unlikely to fully support management

goals [e.g., 16,28,35], and it is becoming increasingly clear that reaches in the arid western United States can differ substantially in their flow regimes. For example, the first split in our hierarchical cluster analysis distinguished two classes, one of which (B = 2/3rd of the reaches) contained both nonperennial and perennial reaches. The other class was wholly nonperennial (A). Class A was subsequently split into two nonperennial classes (A1 and A2, Figure 3) based largely on number of zero-flow days, the magnitude of peak flows, and presence of baseflow. These two A classes appear to be similar to two classes identified by McManamay and DeRolph [43], who developed a flow-regime classification that spanned the entire conterminous USA. The distinction between these two classes may be critical from a management perspective because states like Arizona must identify ephemeral reaches from other nonperennial reaches for regulatory purposes. It is unclear, however, how tightly our ephemeral class (A1) matches definitions of ephemeral used by different regulatory entities. These details notwithstanding, the identification of different types of nonperennial reaches as shown here will likely be important in the development and application of consistent regulatory policies designed to protect regulated waterbodies. Our modeling and mapping of reach classes (Figure 7) is a step in that direction, especially given that the prediction error of 16% for the most ephemeral (A1) reaches should represent a reasonable level of uncertainty for at least initial screening purposes. Use of higher-resolution DEMs and stream networks may also improve both classification and predictions. Higher-resolution DEMs are becoming increasingly available [74], although comparable higher-resolution digital stream networks are not yet widely available.

The presence of permanent flow, or its lack thereof, is one aspect of the flow regime, and it may not always be obvious how it should be weighted when classifying flow regimes. From the cluster analysis, pure perennial reach classes were unresolved until the 4th-level (five classes) of the dendrogram, i.e., one of five classes (B1b). Further splits in the cluster diagram were needed to resolve additional pure or nearly pure classes of nonperennial and perennial reaches, e.g., the seven-group classification (Table 4), where three of seven classes (B1b.i, B1b.ii, B2a) were perennial and four of seven classes (A1, A2, B1a, B2b) contained 90–100% nonperennial reaches (Figure 3). However, the last classes formed, nonperennial B2a and perennial B2b, had unreasonably high prediction errors, which greatly compromises their potential utility. These two classes had highly similar overall flow regimes (Figure 5) other than the presence (B2a) and absence (B2b) of zero flow, and these two classes had a high chance of being misclassified as the other class (Table 4). The overall similarities in their flow regimes could result in high rates of misclassification and hence a failure to distinguish reaches that may be biologically different [18–20], which would lessen the effectiveness of many bioassessment methods [21,79].

We must also emphasize that the classifications that emerged from our analyses are dependent on the specific flow metrics we used to create classes. The metrics used in the hierarchical classification and the thresholds used to define ZFD and ZFY classes were guided by our general ecological knowledge of the importance of flow regimes to aquatic life [19,41]. For example, zero-flow events represent severe disturbances that can reset biological communities, and the number, timing, and duration of such events should influence the magnitude and predictability of the recovery dynamics of stream communities [4,6,17,22]. We also point out that zero-flow measures at a gauge do not necessarily indicate that an entire reach is dry—pools may still be present that can provide important refugia to aquatic biota [17]. However, until these classifications are tested to assess whether they are actually associated with variation in valued ecological attributes, we do not yet know that the types of flow regimes we identified will be useful or how many classes will be needed. Managers ultimately need classifications that allow enough partitioning of biological variation to produce indices that are precise enough to detect ecologically important impairment but do not have so many classes that they are difficult to communicate and unwieldy to use. Assessing the strength of these relationships is an active area of research [4,6,10,28,79]. Use of a different set of flow metrics would likely

lead to different types and numbers of classes [10], and it is possible that such alternative classifications might be as, or more, useful than the ones developed here.

### 4.3. Predictive Variables and Prediction Errors

Ideally, predictive models will both perform well (accurate and precise) and be interpretable in terms of what physicoclimatic factors are most important in producing different flow regimes. In this study, both overall and class-specific prediction error rates varied considerably. We observed an expected tradeoff between number of classes and prediction error, which can be used to inform recommendations regarding the practical use of specific classification schemes. If we assume that 25% error is tolerable to water resource managers, our results imply that no more than four metric-based classes will be useful to managers and that none of the zero-flow based classifications would be acceptable. Moreover, in the four-group metric-based classification, the fact that only two classes were predicted with <25% error implies that classification may be of limited utility. These error rates do not seem to be unique to our study. For example, Dhungel et al. [41] reported error rates that ranged from 14 to 43% for an eight-group classification, with the most error in small non-perennial reaches and the least error in small perennial reaches. In contrast, McManamay and DeRolph [43] report overall prediction errors of ~5 to 34% for classifications based on 2 to 30 hierarchically-defined flow-regime classes across the entire conterminous USA. They did not report class-specific error rates. Their analyses did include three classes of reaches in their 30-group classification that they described as intermittent flashy, but based on their maps, only two of these reach classes appear to exist in the southwestern USA. We expect that the higher error rates that we observed in our study occurred because our classification of southwestern reaches was much more resolved because of the increased coverage and length of nonperennial reach gauge records. It appears that we can predict the ends of the ephemeral to perennial continuum reasonably well, but our models had difficulty distinguishing reaches that are more variable in the number of zero-flow days from year to year. It is also possible that more spatially resolved predictors could improve predictions. We used 30-m DEMs to characterize most of our predictors. LiDAR (Light Detection and Ranging)-based mapping by the USGS with a resolution of 1–3 m is becoming increasingly available, which may improve predictions.

We also point out that a specific reach could be predicted to be in different classes depending on the number of other classes used in the classification (Figure 7). These differences occurred because each model (two-group, three-group, etc.) differed somewhat in class-specific errors, i.e., the classification itself affected modeling error. This observation may be an under-appreciated aspect of flow-regime modeling of which researchers and mangers need to be aware.

Empirical models can perform poorly if predictor variables that represent important drivers of streamflow patterns are missing from the models. We attempted to ensure that the predictor variables we used represented the range of physicoclimatic controls on flow regimes expected to occur in arid regions. The specific variables that the *VSURF* procedure identified as optimal sets of predictors in the models were generally similar across classifications and interpretable in terms of their likely mechanisms. For example, BFI, which represents the slowly moving source of streamflow [38,80,81], was consistently ranked as a top predictor (Table S2, Supplementary Materials), likely due to its clear linkage to the streamflow patterns themselves. However, it may be unlikely that prediction errors can be substantially improved by incorporating additional predictor variables, and we suspect that flow regimes of individual reaches in arid regions may be inherently difficult to predict because physicoclimatic conditions and streamflow generation may not be as tightly coupled in arid regions as they are in more mesic regions. The physical controls on streamflow response in arid regions are typically related to intermittent monsoonal climate patterns, low annual precipitation [15,17,39,41], comparatively high annual rates of potential evapotranspiration [45,46,82], and shallow bedrock, leading to lower annual baseflows and often irregular patterns of zero flow [15,34–36]. As a result, rainfall and

runoff are often poorly correlated and have nonlinear dependencies on antecedent conditions [23,55,83]. Aryal et al. [55] found that, similar to other studies in semi-arid regions, streamflow patterns were much more muted than precipitation patterns, the spatial distribution of streamflow did not correlate with the spatial pattern of precipitation, and there was no significant correlation between the annual number of zero-flow days and days with precipitation, or between catchment area and mean annual or seasonal runoff.

## 5. Conclusions

Ultimately, any useful classification of streamflow regimes must either generate new understanding of the processes that influence stream flows or help managers identify reaches that differ with respect to their management objectives. Our study focused on developing classifications of streamflow regimes that would allow resource managers to more fully characterize the diversity of flow regimes that occur in arid landscapes. For example, the Arizona Department of Environmental Quality (ADEQ, Phoenix, Arizona, USA) has estimated that approximately 90% of stream length across Arizona may no longer be considered as Waters of the United States (WOTUS) under the USEPA Navigable Waters Protection Rule of 2020 and hence will not be protected under the Clean Water Act (personal communication, Patrice Spindler, ADEQ). This new rule defines ephemeral waterbodies as nonjurisdictional under the Clean Water Act, which affects Arizona's universe of WOTUS and all water quality management programs. In Arizona, flow-regime map updates at the state level and to the NHD have been ongoing; however, flow data are sparse for ephemeral and intermittent reaches, so updates to these waterbodies is incomplete at best and grossly inaccurate at worst [35]. The flow-regime modeling described here provides a tool for identifying where ephemeral and other types of reaches likely occur, when other data are lacking. These flow-regime data will help with WOTUS determinations regardless of what specific criteria are used to define WOTUS, and they should provide another tool for identifying flow regime, designated uses, and appropriate standards applications.

Classifications that extend beyond the traditional dichotomy of perennial and non-perennial reaches are almost certainly needed to support developing programs tasked with assessing whether reaches are meeting physical, chemical, and biological water quality standards. At a minimum, our study showed that it is possible to distinguish a subset of nonperennial reaches that are ephemeral and likely differ biologically from both other nonperennial and perennial reaches. An equally important goal is to better understand what level of resolution in flow-regime classifications is needed to best support developing bioassessment programs. Our study adds to a growing body of knowledge that is needed to address that question for stream ecosystems in the arid southwestern USA and elsewhere. We also need to recognize that climate change is causing flow regimes to change in the arid southwestern USA and elsewhere [84]. An important assumption when developing flow-regime classifications for use in bioassessment is that the data used to create the classifications will adequately represent future flow regimes. Climate has changed in the southwestern USA in recent decades, and we observed drying trends in many of the flow records that we compiled. How to parse the effects of natural variation in flow regimes on stream ecosystems from those caused by climate change represents a major research challenge moving forward.

**Supplementary Materials:** The following are available online at https://www.mdpi.com/2073-4441/13/3/380/s1, Table S1: List of 95 watershed, catchments, and streambed profile attributes mined and calculated from available sources. Table S2: Lists of predictors (with source) that were selected by the *VSURF* R package for the five hierarchical classification models, the four zero-flow day and zero-flow year models, and the continuous zero-flow days and zero-flow years regression models.

**Author Contributions:** Conceptualization, A.M.M. and C.P.H.; data curation, A.M.M.; formal analysis, A.M.M.; funding acquisition, C.P.H.; investigation, A.M.M. and C.P.H.; methodology, A.M.M., B.L. and C.P.H.; project administration, C.P.H.; resources, C.P.H.; software, B.L.; supervision, B.L. and C.P.H.; visualization, A.M.M., B.L. and C.P.H.; writing—original draft, A.M.M. and C.P.H.;

writing—review and editing, B.L. and C.P.H. All authors have read and agreed to the published version of the manuscript.

**Funding:** This research was funded by the Great Lakes Environmental Center, Traverse City, MI, USA, contract number 16583.

**Institutional Review Board Statement:** Not applicable.

**Informed Consent Statement:** Not applicable.

**Data Availability Statement:** The data used in this study are openly available in USGS Water Mission Area NSDI Node, USGS National Water Information System: Web Interface, EPA Watershed Assessment, Tracking & Environmental Results System, NASA National Snow and Ice Data Center Distributed Active Archive Center, National Drought Mitigation Center, and USGS Earth Explorer. The derived data created in this study are available on request from the corresponding author.

**Acknowledgments:** We thank James Eddings for help running models and preparing the maps of Arizona and Donald Benkendorf for countless hours of advice. We also thank Peter Wilcock for concept guidance. The eflows team at the University of California at Davis, Soluchan Dhungel, Christian Perry, Adam Fisher, Brennan Bean, Emily Burchfield, and Ryan Hill aided in several key aspects of coding.

**Conflicts of Interest:** The authors have no conflicts of interest.

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
