# Peer review of "Classification and Prediction of Natural Streamflow Regimes in Arid Regions of the USA"

_water, doi:10.3390/w13030380_

Round 1

Reviewer 1 Report

The paper of Merritt et al. uses the Random Forest (RF) method with 287 long-term records of streamflow in the south-western USA to develop a series of increasingly detailed classification methods of natural streamflow regimes. Then, the authors evaluate the potential of RF models to predict flow regime class for ungauged reaches and propose a kind of operational output through flow regime maps realized for the state of Arizona.  

In my opinion, the most relevant contribution of this paper is the proposal of a comprehensive methodology (though it is composed of rather well-known tools) to deal with the topic of streamflow regime classification for nonperennial streams. Particularly interesting is the work made to “save” uncompleted records. Results are honestly presented, showing that there is still a long way to go.

My main suggestion for authors is to simplify the manuscript, trying to go more to the point concerning some aspects and summarizing a too long (in my opinion, of course) discussion section. Specifically, I suggest providing a more straightforward physical interpretation of the flow regimes emerging from different groups/classes. This point is hinted at in different sections of the paper, but I would prefer a more explicit explanation, e.g. at the end of paragraph 3.1. Also, the information about top predictors (paragraph 3.6) could be represented with a graph. Furthermore, some choices concerning methodology that could be felt like a bit subjective (e.g., LL271-275) should be explained with some more detail.

Please find below a list of minor corrections and typos. I hope my review can help to improve the quality of the paper.

Minor corrections and typos:

L5: Please check affiliations and numbers referring to authors names, they are not linked correctly.

L32: why “patterns” with capital letters?

L115: I would move “(n=90)” after the word “record” for the sake of clarity

L156: “ten or more years of missing data”. Is there an upper threshold of missing data used to discard a series? In other words, what’s the shortest time series analysed? Please add such information in the paper

L168: please check this sentence

L169: “another”: maybe “other”?

LL176-183: the assumption that the overall percentage of zero-flow days is the same of that found in the raw data for partial streamflow records is debatable and needs to be at least discussed

Eq. (2): not clear how this equation is achieved

L297: what does NHD mean? Furthermore, the definition of local catchment is not so clear. What do the authors mean with “reach”? How was it identified? Based on existing cartography? At what resolution?

L382: “then”: please check

L398: Table S2 is introduced abruptly here. Tables S1 and S2 look pretty the same, please check accurately.

L503: maybe Figure 5?

L572: Table S3 does not exist

Reviewer 2 Report

This is an interesting and important piece of work which explores classification systems for streams and highlights advances in ephemeral systems. While I do not consider much of the methods presented in this work as novel or cutting edge, they add an important contribution to the conversation at a time this is likely to be critical. While I question some aspects of the methodology, I think that some of the limitations are however explained and explored in the discussion. This is however somewhat limited in the aspects covered- there are others. While management is discussed as a central goal of this work, I cannot see its current uptake in this context due to lack of robustness in the method. In saying this however, I see an important longer-term goal for this work to act as a stepping stone for achieving this. This context should be clearly articulated and I think would improve the MS. Some additional context provided in the conclusions would add to the introduction. This is important work and I think this adds to the conversation and likely to well received by researchers in this field.

Please find below more detailed comments provided to improve the manuscript.

30-31 The opening sentence fails to hold true to me. While stream flow characterisation is one possible pathway to assist, it is not however a prerequisite to achieve this.

32- why the capitalisation of ‘patterns’?

33- Again, there are a range of methods that don’t use reference streamflow characterisation.

39- ‘different reference states’ has not been defined. I assume that this is related to condition?

43- state how many years rather than ‘some defined number’

63- mountainous arid regions with nonperennial streams that experience snow melt is an artefact of the stream’s climatic region, not its perenniality.

88- provide some examples of the watershed attributes that have been used in the past

92- presumably you are not using a linear model between rainfall and runoff anyway?

118- Random Forest models. Surely a process based or rainfall-runoff model would be better?

Figure 2 is helpful- with a logical workflow sequence

155- So… 197 streams had 10 or more years of missing data. Does that mean that the 41 ‘continuous’ record streams had up to 10 years of missing data?

166- what are the covariates for the R2 values? It is not clear to me what the inputs are for these models and how the coefficient of determination is calculated (comparing what to what, and at what timestep?)

178- This is where a process-based model would be better. Basically, as I understand what you have done, here you are telling your model to generate the same number of zeros as the available data. This could be wildly inaccurate or biased. This goes beyond correcting sub-zero records and is artificially correcting your generated model to replicate the observed metrics.

Table 2. How is ‘bank full flow’ calculated? Is this based upon observed flows and channel capacity or quantitative methods of analysis of the flow regime?

215- Where these four ‘alternate classifications’ added to the other 7 classes? Or streams that fit these thresholds taken out of the original 7 and reassigned to one of these four? If so, see comments regarding line 178. If I understand this, your model endpoint will contain the same flow characteristics as the partial flow records- this then becomes a very long-winded way of applying these characteristics to steam classes.

 Section 2.7 through 2.9- nice. Although my preference would have been reaches, as my thoughts are this would better enable nesting of different classes within other classes as relevant to the segments. The way that you have done this may be easier to visualise at the appropriate scale however-?

Figure 3 would be improved to incorporate the details occurring in the text above the figure.

Figure 4 would be improved with better axis labels.

Figure 5 would be improved by indicating how many streams fall into each class (that are composites of each panel).

352- Could you better articulate this and what the implications of this are. I think that this will have important ramifications for the interpretation of your method and its results.

357- Why put forward multiple classifications? If the purpose of this is to support management, how is a manager supposed to use this?

Table 4- consider putting this in as appendix

Figure 7- Presumably the area of catchment upstream should be strongly influential in driving the classification of a stream. These maps seem to have less to do with hydrology and more to do with climate- but maybe I am unfamiliar with the region.

430- I would think that much of the first paragraph of the discussion is either opinion, dubious, or otherwise not addressed or improved by the current body of research.

441- I would suggest that a classification system is a very course tool to use in identification of reference condition. A scenario analysis comparing ‘current’ flow conditions to ‘natural’ (without development) would be beneficial for identifying where the resolution of the classification is sufficient for capturing where change has occurred. Hydrological modelling could be used to support this. Looking at the maps in Figure 7, I would suspect that differences in the classification method selected would likely overpower the detection ability of reference compared to current conditions. Understanding this would be critical before applying in a management setting.

  1. Classification in general need not only identify reference condition classes.

Section 4.1 Yes- but there are more robust methods for creating synthetic flow time series than what has been done here.

486- Matching (or checking of the matching) of classes against regulatory entities would be critical.

544- Is this a 25% class assignment error that you are talking about. So this means that one in four streams would be a class type other than the one specified? It would be interesting to know where this error comes from, e.g. creation of synthetic time series, selection of metrics, RF model used etc.

561- generally I think that this paragraph raises an important point. Even the seed of many classification models can change the results. You have opted to show multiple different classifications , while many researchers do not. A critical question is how managers can use these multiple perceptions of the stream network. Literature exists which rationalise how to select these considering an application for management.

 588- I consider that there are other determinates of usefulness than these.

592- Looking at this from an international perspective, the imperative of your classifications to match legislative definitions becomes even more critical.

596- This is interesting (and disheartening). I think that some of this context would be well placed in the introduction.
